# Training load and match-play demands in basketball based on competition level: A systematic review

**Adam J. Petway**[1]*, **Tomás T. Freitas**[2], **Julio Calleja-González**[3], **Daniel Medina Leal**[1], **Pedro E. Alcaraz**[2,4]

1 Philadelphia 76ers Athlete Care Department, Philadelphia, Pennsylvania, United States of America,
2 UCAM Research Center for High Performance Sport, Catholic University of Murcia, Murcia, Spain,
3 Laboratory for Sport Performance Analysis, University of Basque Country, Vitoria, Spain, 4 Faculty of Sport Sciences, UCAM, Catholic University of Murcia, Murcia, Spain

* adampetway@76ers.com

**Data Availability Statement:** All relevant data are within the manuscript and Supporting Information files.

## Abstract

Basketball is a court-based team-sport that requires a broad array of demands (physiological, mechanical, technical, tactical) in training and competition which makes it important for practitioners to understand the stress imposed on the basketball player during practice and match-play. Therefore, the main aim of the present systematic review is to investigate the training and match-play demands of basketball in elite, sub-elite, and youth competition. A search of five electronic databases (PubMed, SportDiscus, Web of Science, SCOPUS, and Cochrane) was conducted until December 20th, 2019. Articles were included if the study: (i) was published in English; (ii) contained internal or external load variables from basketball training and/or competition; and (iii) reported physiological or metabolic demands of competition or practice. Additionally, studies were classified according to the type of study participants into elite (20), sub-elite (9), and youth (6). A total of 35 articles were included in the systematic review. Results indicate that higher-level players seem to be more efficient while moving on-court. When compared to sub-elite and youth, elite players cover less distance at lower average velocities and with lower maximal and average heart rate during competition. However, elite-level players have a greater bandwidth to express higher velocity movements. From the present systematic review, it seems that additional investigation on this topic is warranted before a "clear picture" can be drawn concerning the acceleration and deceleration demands of training and competition. It is necessary to accurately and systematically assess competition demands to provide appropriate training strategies that resemble match-play.

## 1. Introduction

Basketball is a court-based team sport that requires proficiency in a vast array of physical parameters and motor abilities (i.e., speed, strength, and endurance) to achieve success from both a technical and tactical standpoint [1]. The ability to accelerate, decelerate, change direction, jump, and shuffle are paramount for on-court success, due to the intermittent high-

**Funding:** The authors received no specific funding for this work.

**Competing interests:** The authors have declared that no competing interests exist.

intensity nature of most actions and basketball-specific movements [2,3] as well as the demands of the sporting activity [4,5,6]. Importantly, in competition settings, the aforementioned abilities must be expressed in an efficient and economical manner over the course of four quarters with contributions from both aerobic and anaerobic energy pathways [1]. In this context, the density of game-related activity (determined by specific work-to-rest ratios) is dictated by action intensity and by the moment of the game [7]. This includes medium- to high-intensity actions that last 15 seconds (s) and high- to maximal-intensity actions that last up to 2–5 s [8,9]. It is for this reason that practitioners must have a precise overview of match-play demands as well as the load elicited during training [4,5,2,6,10,3,11,12,13,14,15]. In fact, over the past years, there have been several studies documenting match-play demands in basketball [4,5,2,6,10,3,11,12,13,14,15,16,7,17,18,19,20,21,22,23,24,9,25,26,27,28]. Particularly, a recent review by Stojanovic et al. [29] analyzed the activity demands and physiological responses obtained during basketball competition and found that playing period, playing position, level, geographical location and sex greatly influenced the stress experienced by basketball players. In their article Stojanovic et al. [29] examined heart rate (HR), blood lactate concentration, total distance, and movement patterns of male and female basketball competitions based on time-motion analysis. However, while the study clearly described the competition characteristics, the authors did not present data on the acceleration/deceleration requirements of the game nor did they examine the demands of training versus match-play. It is for these reasons that the current systematic review is justified.

It is important to note that amongst the several methods used to quantify the demands of play, and regarding internal load quantifications, HR [6,3,11,12,14,20] and blood lactate concentration [4,13,14,16,9,30] were the most frequently used. In fact, internal variables such as average and maximal HR can be extracted to quantify loading parameters during match-play [11,12,21,30,26]. Concerning external load, methods such as accelerometry and the use of positional tracking cameras [4,2,13,16,7,17,31] are amongst the most common. Within this framework, total or high-intensity accelerations and decelerations, total distance traveled, and top speed reached were the widely used variables to assign a value to the mechanical load imposed. In addition, time-motion analysis [4,14,18,22,9,26,32] measuring time and frequency of movements such as "standing"; "jogging"; "running"; "sprinting"; and "jumping" during competition can be found in the literature. Despite match-play demands based on time-motion analysis having been found to present a high level of variability according to playing position, skill level and training age [29], no robust evidence exists regarding the use of accelerometry. Therefore, a systematic analysis of both approaches to match demands quantification is warranted. Collectively, a better understanding of this 'real-time' feedback can give relevant and useful information concerning normative group standards, as well as relative to the individual athlete. Additionally, having a clear "picture" of both internal and external loading parameters can provide a better insight into global stress that the players deal with during training and competition [2,10,26].

In a related topic, tracking training load in this team-sport may be of extreme importance to ensure that the players are physically prepared for competition demands from a fitness standpoint, in order to avoid acute spikes in load from a fatigue and injury prevention perspective [3,11,7,17] and to provide individualized recovery strategies [33,34]. With this in mind, a copious amount of research has also been focused on investigating and describing basketball training load parameters over recent years [35,36,37,38,39,40,41,21,42,24,43,44]. As previously mentioned for competition, accelerometry is becoming an increasingly popular means of quantifying load during training [36,38,40,21]; however, no conclusive data has been reported throughout the different studies. For this reason, a more in-depth and systematic analysis of the literature is warranted. Regarding internal load, HR and session rate of perceived exertion

(sRPE) (i.e., the subjective feedback from the player on a 1–10 scale multiplied by duration of training) have been shown to be a cost-effective way of providing valuable information widely used by coaches and sport scientists [35,37,41]. Remarkably, an important variability has been reported within basketball training loads based on quantification means of training load, position, perceived exertion, skill level, and training age [36,37,38,39,40,41,43,44], once again identifying the need for a systematic review of the published data.

The current state of the literature is not conclusive regarding the typical training load experienced by basketball players of different competition levels given that only match-play demands and physiological responses during competition have been previously described [29]. To our knowledge, no previous investigation has focused on systematically reviewing the literature to identify precise loads during training versus match-play whilst clearly defining different levels of competition. As such, there is an important gap in the available research that does not allow concluding whether basketball training is closely mimicking game demands, hence, adequately preparing players for the stress imposed by competition. Moreover, new technologies that allow quantifying the acceleration/deceleration demands in basketball training and competition have emerged, but no current literature review has addressed this topic. Therefore, the aim of the present systematic review is to analyze the evidence related to the training load and match-play demands of basketball across different levels of competition.

## 2. Materials and methods

### 2.1 Study design

The present study is a systematic review focused on training load and match-play demands at different levels of competition in basketball. The review was not registered prior to initiation, was performed in accordance with the Preferred Reporting Items for Systematic Reviews and Meta Analyses (PRISMA) statement [45] and did not require Institutional Review Board approval.

### 2.2 Search strategy

A structured search was carried out in PubMed, PubMed Central, Web of Science, SportDiscus and Cochrane databases, all high quality databases which guarantees strong bibliographic support. The electronic database search for the related articles considered all publications prior to December 20th, 2019. The following key words were used to conduct the search "basketball", "training load", "accelerometry", "load monitoring", "internal load", "total distance", "average distance", "top speed", "average speed", "metabolic", "heart rate", "competition demands", "training demands", "training", and "rate of perceived exertion". In addition, the key word "basketball" was present in each search to ensure that the relevant information was catered to articles involving only this sport. The reference sections of all identified articles were also examined (by applying the "snowball methods" strategy [40]). Once the electronic search was conducted, relevant studies were identified and organized in a systematic fashion.

All titles and abstracts from the search were cross-referenced to identify duplicates and any potential missing studies, and then screened for a subsequent full-text review. The search for published studies was independently performed by two authors (AP and TTF) and disagreements were resolved through discussion.

### 2.3 Inclusion and exclusion criteria

This review included cross-sectional and longitudinal studies considering healthy, professional or junior, male basketball players. Study participants were categorized into three groups: elite,

sub-elite, and youth. The elite basketball group was defined as teams participating in the NBA, NBA G-League, NCAA Division I, Euro League, FIBA International Competition, ACB, Top Divisions in Europe, South America, Australia, and Asia. Sub-elite was defined as professional or semi-professional that did not meet the elite criteria but were over 19 years old. Youth was considered for studies in which the participants were all 19 years of age or younger. Studies were included in the present review if they met the following criteria: (i) the study was published in English; (ii) the study included internal or external load variables from basketball training and/or competition; and (iii) the study reported physiological or metabolic demands of competition or practice.

Studies were excluded if (i) the study participants were wheelchair basketball players; (ii) the study participants were female; (iii) the data being collected did not describe training load or competition demands; and (iv) the study consisted on a review or a conference proceeding.

## 2.4 Study selection

The initial search was conducted by one researcher (AP). After the removal of duplicates, an intensive review of all of the titles and abstracts obtained were conducted. Following the first screening process, the full-version of the remaining articles was read. Then, on a blind, independent fashion, two reviewers excluded studies not related to the review's topics and determined the studies for inclusion (AP and TTF), according to the criteria previously established. If no agreement was obtained, a third party intervened and settled the dispute. Moreover, PEDro scale (Fig 1) was used to evaluate whether the selected randomized controlled trials were scientifically sound (9–10 = excellent, 6–8 = good, 4–5 = fair, and <4 = poor) [46]. Papers with poor PEDro score were excluded. Final outcomes of the interventions were extracted independently by two authors (AP and TTF) using a customized spreadsheet (Microsoft Excel 2016, USA). Disagreements were resolved through discussion until a consensus was achieved.

## 3. Search results

As several databases were scrutinized, the initial database search yielded 18,805 citations. After duplicate removal, 3,282 abstracts and titles were left for review. Upon screening, 165 articles met the inclusion criteria for full-text review. Of the 165 articles reviewed, 35 met the criteria for the systematic review. Of the 35 articles that met the criteria, 12 had participants for elite competition demands [4,5,6,11,12,13,14,15,16,7,9,30,32], 16 articles had participants for elite training load [2,10,3,12,15,35,37,38,39,41,20,42,25,27,43,47], 6 for sub-elite competition demands [4,11,13,21,26,32], 3 for sub-elite training load [23,44,48], 5 for youth competition demands [11,18,22,9,28] and 1 for youth training load [24]. A full view of the search and selection process can be found in the PRISMA flow diagram [45] in Fig 2.

## 4. Competition demands

### 4.1 Internal competition load

Internal load outcomes pertaining to competition demands can be found in Table 1. The variables displayed in the different studies consisted of HR and blood lactate concentration.

**4.1.1 Heart rate.** Heart Rate (HR) during competition (Table 1) was organized into two categories according to the classification used in the included studies: maximal ($HR_{max}$) and average HR ($HR_{ave}$). The values of $HR_{max}$ during elite level competition ranged from 187 to 198 beats per minute (BPM) with a mean of 190 BPM [11,12,30]. With regards to sub-elite competition, values ranged from 192 to 195 BPM with a mean of 194 BPM [11,21,26]. In addition, in youth competition, the $HR_{max}$ held a mean of 199 BPM [11,18]. The data extracted

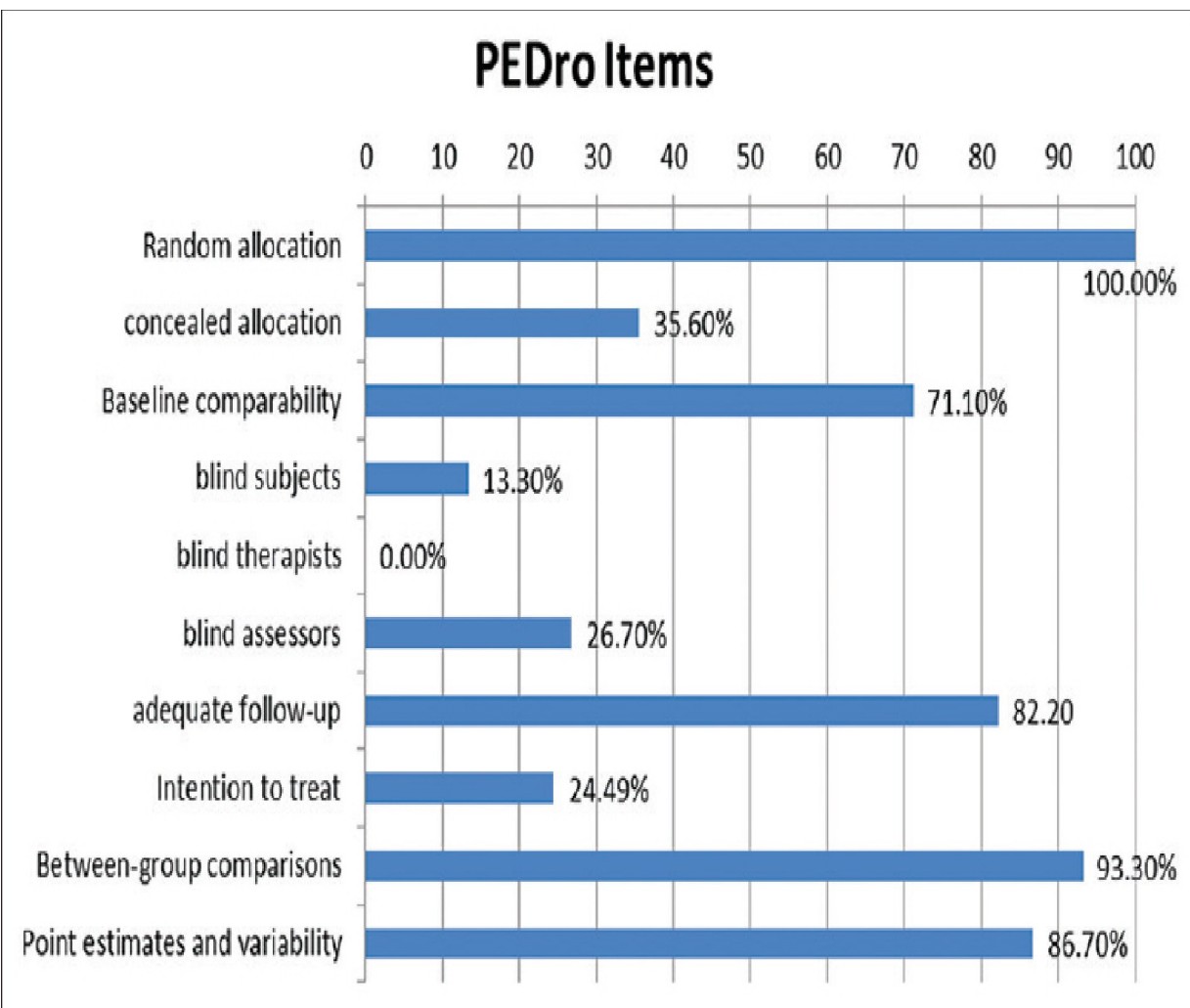

**Fig 1. PEDro scale.**

indicated that elite competitors presented lower $HR_{max}$ values during competition, which can be interpreted as an indicator of elite players having a higher overall level of fitness and a more efficient work rate compared to sub-elite and youth players [11]. Interestingly, according to the results retrieved from the literature, the same pattern occurred with the $HR_{ave.}$ During elite level competition the value ranged from 150 to 175 BPM [11,12,30], in sub-elite competition ranged from 168 to 169 BMP [11,21] and in youth competition the $HR_{ave}$ ranged from 167 to 172 BPM [11,18].

**4.1.2 Blood lactate concentration.** Blood lactate concentration was collected as an internal measurement during select studies of elite level competition. The samples for mean blood lactate post-competition held an average of 5.1 ± 1.3 mmol/L [18,21,9] with a range of 4.2 to 5.7 ± 1.2. Abdelkrim et al. [9] observed a peak of 6.2 ± 1.3 in the fourth quarter for the Tunisian National Team. The fourth quarter peak is likely due to the build-up of blood metabolites and catabolic hormones based on the depletion of muscle glycogen later in competition. The ability to buffer these mechanisms internally may have had a direct impact on mechanical outputs during competition [30] as internal load parameters leading to fatigue have been reported to

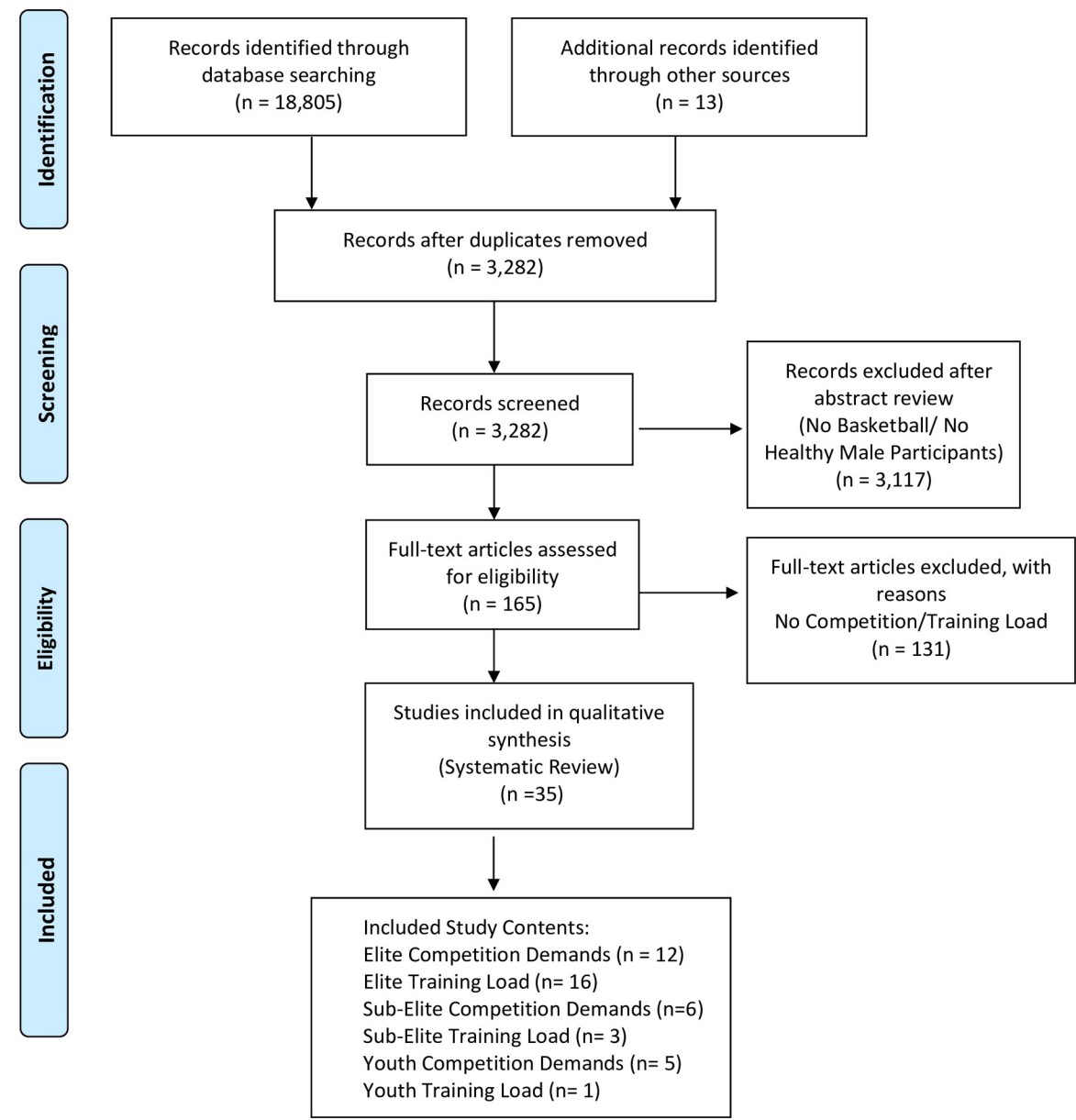

**Fig 2. PRISMA flow diagram.**

negatively affect whole-body work rate, physical and technical performance, and even decision making in team-sports [49]. It is for such a reason that there is a need for future investigation of blood metabolite accumulation during competition and the effects it has on high-speed movement.

## 4.2 External competition load

Table 2 displays the external load variables retrieved from the different studies. Total distance, acceleration (ACC) and deceleration (DEC) efforts during basketball competition, average and top speed reached, and time motion analysis movement frequency and duration were the outcomes extracted.

**Table 1. Internal load during competition.**

| Study | Competitions (n=) | Participants (Competition Level) | % of HR Lactate Threshold | Mean HR (beats/min) | Max HR% | Max HR (beats/min) | Blood Lactate Concentrate (mmol/l) |
|---|---|---|---|---|---|---|---|
| Daniel et al. [6] | n = 6 | Brazilian Basketball League (Elite) | Defense- 104.2 ± 2.21 Offense- 103.7 ± 1.80 Defense Transition- 104.8 ± 2.44 Offense Transition- 104.3 ± 3.55 | | | | |
| Lopez-Laval et al. [11] | n = 3 | Spanish ACB League/ABA/ Spanish Juniors (Elite/Sub-Elite/Youth) | | Elite Adults- 150 ± 11 Amateur Adults- 168 ± 9 Elite Juniors- 167 ± 10 | Elite Adults- 79 ± 4 Amateur Adults- 87 ± 3 Elite Juniors- 84 ± 4 | Elite Adults- 190 ± 2 Amateur Adults- 193 ± 4 Elite Juniors- 199 ± 3 | |
| Abdelkrim et al. [18] | n = 9 | Tunisian U-19 National Team (Youth) | | All Positions- Q1-173 ± 4 Q2-173 ± 5 Q3-173 ± 4 Q4-167 ± 4 Guards- Q1- 176 ± 4 Q2- 176 ± 5 Q3- 176 ± 4 Q4- 167 ± 4 Forwards- Q1- 173 ± 5 Q2- 173 ± 5 Q3- 174 ± 4 Q4- 167 ± 4 Center- Q1- 171 ± 3 Q2- 170 ± 3 Q3- 171 ± 4 Q4- 165 ± 4 | All Positions- 91 ± 2 | | Mean-5.49 ± 1.24 mmol/l |
| Torres-Ronda et al. [12] | n = 7 | Spanish ACB League (Elite) | | 158 ± 10 | 96.8 ± 2.6 | 198 ± 9.3 | |
| Abdelkrim et al. [9] | n = 6 | Tunisian Junior National Team (Youth) | | | | | Mean-5.75 ± 1.25 mmol/L Peak- 6.22 ± 1.34 |
| Abdelkrim et al. [30] | n = 6 | Tunisian National Team (Elite) | | Q1-176 ± 5 Q2-176 ± 4 Q3-176 ± 4 Q4-172 ± 4 | | | |
| Narazaki et al. [21] | n = 1 | NCAA Division II (Sub-Elite) | | 169.3 ± 4.5 | | | 4.2 ± 1.3mmol/L |
| Puente et al. [26] | n = 1 | Spanish Basketball Federation (Sub-Elite) | | | Guards- 89.6 ± 4.7 Forwards- 87.8 ± 3.2 Centers- 92.7 ± 4.7 Whole Group- 89.8 ± 4.4 | | |

Heart Rate (HR) expressed in Beats Per Minute (BPM). Blood Lactate Concentrate express in millimoles per liter mmol/L. Q1 is 1st quarter, Q2 is 2nd quarter, Q3 is 3rd quarter, and Q4 is 4th quarter of match-play.

**Table 2. External load during competition.**

| Study | Competitions (n=) | League (Level) | Average Speed | Max Speed | Total Distance | Accelerations | Decelerations |
|---|---|---|---|---|---|---|---|
| Sampaio et al. [5] | n = 1230 | NBA (Elite) | Speed in offense (m·s) All-Star 1.95 ± 0.16 Non-All-Star 2.01 ± 0.12 Speed in defense (m·s) All-Star 1.63 ± 0.07 Non-All-Star 1.72 ± 0.08 | | | | |
| Scanlan et al. [13] | n = 5 | Australian NBL/ Queensland State Basketball League (Elite/ Sub-Elite) | | | Professional Quarter 1–1653 ± 38 Quarter 2–1591 ± 24 Quarter3-1531 ± 72 Quarter 4–1504 ± 21 Semiprofessional Quarter 1–1549 ± 81 Quarter 2–1601 ± 88 Quarter 3–1501 ± 166 Quarter 4–1557 ± 238 | | |
| Vázquez-Guerrero et al. [27] | n = 2 | Spanish ACB League (Elite) | | | | PGs- Acc. ($<3$ m·s$^{-2}$) #/min- 29.6 ± 3.9 Acc. ($>3$ m·s$^{-2}$) #/min- 1.4 ± .9 SGs- Acc. ($<3$ m·s$^{-2}$) #/min- 32.7 ± 11 Acc. ($>3$ m·s$^{-2}$) #/min- 1 ± .4 SFs- Acc. ($<3$ m·s$^{-2}$) #/min-26.7 ± 2.6 Acc. ($>3$ m·s$^{-2}$) #/min-.8 ± .3 PFs- Acc. ($<3$ m·s$^{-2}$) #/min- 28 ± 5 Acc. ($>3$ m·s$^{-2}$) #/min- 1.4 ± .5 Cs- Acc. ($<3$ m·s$^{-2}$) #/min- 28.3 ± 1.1 Acc. ($>3$ m·s$^{-2}$) #/min- 1.5 ± .4 | PGs- Dec. ($<-3$ m·s$^{-2}$) #/min- 23.8 ± 3.6 Dec. ($>-3$ m·s$^{-2}$) #/min- 4.5 ± 1.4 SGs- Dec. ($<-3$ m·s$^{-2}$) #/min-25.7 ± 10 Dec. ($>3$ m·s$^{-2}$) #/min- 4.5 ± 1.4 SFs- Dec. ($<-3$ m·s$^{-2}$) #/min- 21.7 ± 2.2 Dec. ($>-3$ m·s$^{-2}$) #/min- 3.2 ± .7 PFs- Dec. ($<-3$ m·s$^{-2}$) #/min- 24 ± 4.6 Dec. ($>-3$ m·s$^{-2}$) #/min- 3.5 ± .7 Cs- Dec. ($<-3$ m·s$^{-2}$) #/min- 23.4 ± 1.3 Dec. ($>-3$ m·s$^{-2}$) #/min- 3.7 ± .8 |
| Svilar et al. [15] | n = 11 | Spanish ACB League (Elite) | | | | tACCmin- 2.19 ± 0.84 (2.07–2.31) hACCmin- 0.38 ± 0.25 (0.34–0.42) | tDECmin- 2.38 ± 0.63 (2.28–2.47) hDECmin- 0.25 ± 0.19 (0.22–0.28) |
| Caparrós et al. [7] | n = 87 | NBA (Elite) | | Average- 8.09 ± 0.44 (m·s) Minimum-6.79 (m·s) Maximum-8.76 (m·s) | | Acceleration- .5 (m·s$^{-2}$)-262.5 ± 97.9 1 (m·s$^{-2}$)- 90.2 ± 34 2 (m·s$^{-2}$)- 12.8 ± 34 4 (m·s$^{-2}$)- 0.7 ± 1.0 | Deceleration- -.5 (m·s$^{-2}$)- 172.7 ± 62.7 -1 (m·s$^{-2}$)- 112.3 ± 39.1 -2 (m·s$^{-2}$)- 6.6 ± 3.6 -4 (m·s$^{-2}$)- 0.3 ± 0.6 |

(*Continued*)

**Table 2.** (Continued)

| Study | Competitions (n=) | League (Level) | Average Speed | Max Speed | Total Distance | Accelerations | Decelerations |
|---|---|---|---|---|---|---|---|
| Abdelkrim et al. [16] | n = 6 | Tunisian National Team (Elite) | | Peak Speed- (m·s) PG- 5.2 ± .52 (4.02–5.76) SG4.60 ± 0.42 (4.02–5.29) SF- 4.69 ± 0.63 (4.02–5.76) PF- 4.72 ± 0.61 (4.02–5.76) C- 4.10 ± 0.35 (3.78–4.79) | PG- 2,724 ± 711 (1,120–3,480) SG- 1,907 ± 577 (1,120–2,840) SF- 2,031 ± 867 (1,120–3,480) PF- 2,067 ± 837 (1,120–3,480) C- 1,227 ± 484 (800–2,160) | | |
| Puente et al. [26] | n = 1 | Spanish Basketball Federation (Sub-Elite) | | Max Speed (m·s) Guards- 6.6 ± 0.4 (5.9–7.3) Forwards- Max Speed- 6.2 ± 1.1 (5.1–8.5) Center- Max Speed- 5.9 ± 0.4 (5.1–6.3) Whole group- Max Speed- 6.2 ± 0.7 (5.0–8.5) | | | |
| Abdelkrim et al. [9] | n = 6 | Tunisian National Team (Elite) | | | Total Distance 7,558 ± 575 (6,338–8,397). 1st half- 3,742 ± 304 2nd Half- 3,816 ± 299 m | | |
| Vázquez-Guerrero et al. [28] | n = 13 | Euro League U-18 (Youth) | | Peak Speed (km·h⁻¹) Guards- Q1- 19.57 ± 0.9 Q2- 19.56 ± 1.3 Q3- 19.64 ± 0.8 Q4- 19.36 ± 1.0 Forwards- Q1- 19.35 ± 1.0 Q2- 39.34 ± 1.0 Q3- 18.92 ± 0.3 Q4- 19.15 ± 1.0 Center- Q1- 19.16 ± 0.8 Q2- 18.82 ± 1.0 Q3- 18.75 ± 1.0 Q4- 19.07 ± 0.9 | Total Distance/Playing Duration Guards- Q1- 80.46 ± 7.5 Q2- 73.91 ± 8.9 Q3- 76.81 ± 8.4 Q4- 70.00 ± 9.8 Forwards- Q1- 78.91 ± 10.0 Q2- 71.90 ± 9.0 Q3- 71.98 ± 11.2 Q4- 69.15 ± 13.8 Centers- Q1- 73.45 ± 12.9 Q2- 69.10 ± 7.9 Q3- 68.95 ± 9.4 Q4- 64.24 ± 8.5 | Acc. > 2 (m·s⁻¹) Guards- Q1- 2.20 ± 0.4 Q2- 1.99 ± 0.6 Q3- 1.95 ± 0.5 Q4- 1.72 ± 0.4 Forwards- Q1- 2.04 ± 0.6 Q2- 1.83 ± 0.5 Q3- 1.72 ± 0.5 Q4- 1.66 ± 0.6 Centers- Q1- 1.76 ± 0.6 Q2- 1.64 ± 0.4 Q3- 1.44 ± 0.3 Q4- 1.26 ± 0.4 | Dec. > -2 (m·s⁻¹) Guards- Q1- 2.04 ± 0.4 Q2- 1.79 ± 0.5 Q3- 1.82 ± 0.5 Q4- 1.52 ± 0.4 Forwards- Q1- 1.70 ± 0.5 Q2- 1.47 ± 0.5 Q3- 1.39 ± 0.5 Q4- 1.28 ± 0.5 Centers- Q1- 1.25 ± 0.4 Q2- 1.20 ± 0.4 Q3- 1.04 ± 0.3 Q4- 0.99 ± 0.4 |

(m·s) = meters per second. (km·h) = kilometers per hour PG- Point Guard, SG-Shooting Guard, SF- Small Forward, C- Center. Acc. = accelerations. Dec. = decelerations. tACC = total accelerations. hACC = high-intensity accelerations. tDEC = total decelerations. hDEC = high-intensity decertations. #/min = number per minute. Q1 = 1st Quarter. Q2 = 2nd Quarter. Q3 = 3rd Quarter. Q4 = 4th Quarter.

**4.2.1 Total distance.** In elite competition, distance traveled ranged from 1,991 to 6,310 m [13,16,9]. The total distance covered during sub-elite competition ranged from 3,722 to 6,208 m [48,13]. Finally, considering youth competition, only one study tracked the distance traveled during competition and reported a value of 7,558 m [9]. Remarkably, there was a discrepancy in distance covered between elite, sub-elite, and youth athletes. Upon review, the elite level basketball athletes covered, on average, less distance (4,369 m) [4,13,16,7], compared to sub-elite

(5,377 m) [4,13,48] and youth players (7,558 m) [9]. This seemingly paradoxical finding suggests that the total distance covered may be a poor indicator of in-game performance. In fact, one could infer that the observed phenomenon is a product of technical mastery relative to the demands of competition, as well as elite level players having a higher level of economy in relation to the tactical aspects of basketball [1,5,6]. Based on the present results and as it occurs in other team-sports [50], the key aspect here appears to be not "*how much*" distance a player covers (i.e., quantity) but "*how*" and at "*what intensity*" that distance is covered (i.e., quality). In fact, in support of the previous, Sampaio et al., [5] suggested that better players tend to make fewer mistakes when deciding when and where to run which may result in shorter paths to reach their destination. This is more than likely due to a high degree of technical and tactical discipline based on training age and experience, more hours of professional supervised practices, and higher level of coaching.

**4.2.2 Accelerations and decelerations.**   Accelerometry in basketball is tracked via inertial units containing accelerometer, gyroscope, and magnetometer sensors [15,7,27]. These sensors allowed inertial movement analysis by recording accelerations, decelerations, jumps, and changes of direction (COD). As it can be seen in Table 2, when considering the accelerometry data collected during elite level competition, most research breaks it down into two important categories: accelerations (ACC) and decelerations (DEC) [15,7,27,28]. Additionally, two subsections of these categories can be found: total (T), and high-intensity (HI) [15,27]. For the purpose of this review, total accelerations ($ACC_T$) were classified as total forward acceleration, whereas high-intensity accelerations ($ACC_{HI}$) were classified as the total forward acceleration within the high band ($>3.5$ m·s$^{-2}$) [15], and ($>3$ m·s$^{-2}$) [27]. Total decelerations ($DEC_T$) consisted of the total number of decelerations and high-intensity decelerations ($DEC_{HI}$) were classified as total deceleration within the high band ($>-3.5$ m·s$^{-2}$), and ($>-3$ m·s$^{-2}$) [27].

During elite level match-play, the $ACC_T$ ranged from 43 to 145, and the total number of $ACC_{HI}$ ranged from 1 to 15 per match. Remarkably, a substantial variability can be found within the included studies, considering the ACC values. This occurrence makes it difficult to draw precise conclusions regarding the ACC demands of elite basketball competition. In fact, a similar pattern can be observed for $DEC_T$ as values ranging from 24 to 95 per match were found. Regarding the total number of $DEC_{HI}$ per match, data extracted ranged from 4 to 40. It seems evident that additional investigations on this topic are warranted before a "clear picture" can be drawn concerning the ACC and DEC demands. Moreover, researchers and sports scientists are encouraged to follow a standardized approach to ACC and DEC quantifications (e.g., determining the same HI bands) so that comparisons between studies and data sets can be conducted. None of the sub-elite or youth teams in the included studies collected accelerometry data during competition.

**4.2.3 Average and top speed.**   Studies evaluating NBA competition [5,7] recorded average speed in miles per hour (mph), but values were converted by the authors to the global unit measurement of meters per second (m·s$^{-1}$). The speed recorded by using spatial tracking cameras (Sport VU$^®$; Chicago, USA) can be seen in Table 2. Sport VU$^®$ cameras were installed in all 30 NBA arenas from the 2012–2013 season until the 2016–2017 season and McLean et al. [51] collected data from the entire 82 games plus the playoffs. This technology uses computer vision systems designed with algorithms to measure player positions at a sampling rate of 25 frames per second [5]. Top speed was also measured by Puente [26] via SPI PRO X (GPSports$^®$, Australia) and Abdelkrim et al. [16], as well as Vázquez-Guerrero et al.[28] via WIMU PRO Local Positioning System (Realtrack System, Almeria, Spain).

Similar to accelerometry data, positional tracking cameras have only been used to track match demands in elite level basketball, most likely due to the financial limitations on the sub-elite and youth levels. Importantly, when examining normative data points related to

movements associated with basketball, it seems that the best performers on an elite level expressed certain performance characteristics. For example, Sampaio et al. [5], when examining All-Star Players versus Non-All-Star players in the NBA, found that there was a significant difference in average speed on both the offensive and defensive ends of the court. All-Star players had an average speed of $4.38 \pm 0.36$ mph ($2.0 \pm 0.2$ m·s$^{-1}$) offensively and $3.65 \pm 0.16$ mph ($1.6 \pm 0.1$ m·s$^{-1}$) defensively, whereas Non-All-Star players had an average speed of $4.50 \pm 0.28$ mph ($2.0 \pm 0.1$ m·s$^{-1}$) offensively and $3.86 \pm 0.20$ mph ($1.7 \pm 0.1$ m·s$^{-1}$) defensively. Within the most prestigious level of basketball, the evidence suggests that the most efficient players tend to exert the least amount of energy to achieve the most productive results [5,7]. With regards to top speed, there was also variability among levels. Puente et al. [26] showed that the average top speed in sub-elite Spanish basketball competition was 6.2 m·s$^{-1}$, which is lower than the 8.09 m·s$^{-1}$ average top speed by NBA players identified in the work of Caparrós et al. [7]. However, the former study [26] only analyzed one single sub-elite game and, therefore, caution is warranted when directly comparing the results. For this reason, future research is needed in this area. Taken together, the distance and speed data extracted from the literature hint that higher level basketball players seem to cover less distance but achieve greater top speeds during competition, which is in line with what has been reported in other team sports [52,50].

**4.2.4 Time motion analysis.** Time motion analysis has been widely used to track frequency and duration of movements during competition [4,18,26,22,14,9,32]. Movements such as stand/walk, jog, run, sprint, and jump are commonly recorded among different levels of competition as well as different positions. Within this research, and based on the published literature, stand/walk was defined as movements performed at a velocity of 0–1 m·s$^{-1}$ [1,14,18,22,32] and jogging was defined as intensities greater than walking but without urgency performed at 1.1–3.0 m·s$^{-1}$ [4,18,26,9]. Running was defined as sagittal plane movement at a greater intensity than jogging and with a moderate degree of urgency at 3.1–7.0 m·s$^{-1}$ [18,22,33]. Finally, sprinting was defined as forward movements characterized as effort close to maximum >7.0 m·s$^{-1}$ [4,14,18,9,26,32].

Ferioli et al. [32] and Scanlan et al. [4] examined time motion analysis among elite and sub-elite populations. Upon review, Ferioli et al. [32] found that there was a stark difference between time spent and frequency in high-speed running and sprinting versus jogging in the first division compared to the second division. The 1$^{st}$ Italian Division had frequency of exposures to high-intensity actions (HIA) of $107 \pm 26$, compared to an average of $78 \pm 35$ HIA in the second division. Scanlan et al. [4] found that elite backcourt (EBC) and elite frontcourt (EFC) had a much higher frequency of running compared to sub-elite backcourt (SEBC) and sub-elite front court (SEFC) during match-play. EBC had a mean frequency of $504 \pm 38$ and EFC had a mean frequency of $513 \pm 26$ of exposures to running during competition. These figures for running during competition are much higher than the SEBC ($321 \pm 75$) and SEFC ($352 \pm 25$), respectively. Again, these results would suggest that top-level basketball players spend more time at high-intensity activities compared to their sub-elite counterparts. In addition, elite players tend to display greater control over the most appropriate time and situations to express high-intensity actions relative to the total distance covered whilst on the court.

Abdelkrim et al. [18] and Puente et al. [26] examined the positional differences using time motion variables during competition. Both studies showed that guards spend more time running compared to forwards and centers. Abdelkrim et al. [18] found that guards had a greater frequency of running during competition ($103 \pm 11$), compared to forwards ($88 \pm 5$) and centers ($101 \pm 19$). Puente et al. [26] found that guards run a longer distance of $3.1 \pm 1.1$ (m.min$^{-1}$) compared to forwards ($2.2 \pm 1.9$) and centers ($1.6 \pm 1.6$). This information, seen in Table 3, is useful and may have important implications when prescribing high-intensity running relative to each position in basketball. Based on these results, individual conditioning programs

should be adapted to the specific physical requirements of guards, forwards, and centers, keeping in mind that the latter have been found to have a lower proportion of high-intensity running, acceleration, decelerations, and COD.

## 5. Training demands

### 5.1 Internal training demands

Internal Training Load, displayed in Table 4, considered the following variables: s-RPE, Weekly Training Load, $HR_{max}$, $HR_{ave}$, % $HR_{max}$, and Training Impulse (TRIMP).

**5.1.1 Heart rate.** Heart rate in training was used to quantify the cardiovascular demands imposed on the athletes [3,12,35,20,23,24]. Torres-Ronda et al. [12] examined $HR_{max}$, $HR_{ave}$, and %$HR_{max}$ in 5vs5, 4vs4, 3vs3, 2vs2, and 1vs1 games and found the 1vs1 situations had elicited the largest physiological response. Gocentas et al. [23] compared the $HR_{max}$ between guards and forwards in different training sessions and found that on average guards had a higher HR response (194 ± 14) than forwards (190 ± 12.7). More investigation is needed in the future as it relates to the HR demands of varying training programs.

**5.1.2 Session RPE and total weekly training load.** A fairly common strategy to monitor players' load is to track the total weekly load via the sRPE (RPE multiplied by session duration), collected throughout the training week. In basketball, this method has been widely used to assess Training Load [35, 37, 41] and has been shown to provide good insight on the energy cost of different movement patterns, particularly when coupled with external load data [2,10,39]. Briefly, it involves players reporting their RPE score using the Borg 10-point scale thirty minutes after the completion of each training session, multiplying the value by the number of minutes of the session [41] and then calculating the sum of the values of each training session during the week.

As noted in Table 4, the Total Weekly Training Loads in the studies analyzed ranged from 2255 to 5058 AU in elite level teams [35,37,41]. The large range observed is likely due to the high variability on the number of training sessions or practice duration based on the loads provided by the technical staff. Since sRPE is obtained by multiplying RPE by session duration, the accumulative amount of weekly training load is dependent on the duration of each training session, which can vary based on style of play, level of competition, or moment of the season [36,42,44]. In addition, Svilar et al. [2] found that sRPE showed a very strong correlation with $DEC_T$ and $COD_T$. According to the authors, the rapid eccentric actions involved in decelerations, cuts, and COD may explain the abovementioned relationship [1,2]. Nevertheless, the mechanical stress imposed on the athletes during these movements, as well as the effects of eccentric training in basketball athletes, are areas that need additional investigation in upcoming studies. A key aspect to consider when utilizing this method to monitor training loads and demands is that in the examination of coach and player perception of recovery and exertion, research has shown that coaches tend to overestimate recovery when compared to the athletes' perception [17]. Therefore, when designing appropriate training sessions, a combination of internal and external load variables is recommended [2,10,39].

### 5.2 External training load

Regarding External Training Load (Table 5), the variables retrieved from the studies were the number of ACC, DEC, and COD, tracked with inertial units through accelerometry.

**5.2.1 Accelerations and decelerations.** In elite level basketball, $ACC_T$ in training varied from 16.9 to 59.5 [2,10,15,26,47]. The $ACC_{HI}$ in elite training, classified as the total forward acceleration within the high band (>3.5 m·s⁻²), ranged from 1.9 to 7.2 with a mean of 5.56 per training session. The $DEC_T$ in elite basketball training ranged from 16.4 to 93.2 with a mean of

**Table 3. Frequency, duration, and distance of time-motion analysis during competition.**

| Study | Participants (Competition Level)n = # of comp. | Stand/Walk | Jog | Run | Sprint | Jump | All Movements |
|---|---|---|---|---|---|---|---|
| Scanlan et al. [4] | Australian NBL/ Queensland State Basketball League (Elite/ Sub-Elite) n = 5 | Mean Frequency — EBC- 764 ± 86 SEBC- 462 ± 74 EFC- 815 ± 45 SEFC- 532 ± 38 Duration— mean/total EBC- 0.91± 0.09/691 ±35 SEBC- 2.13 ± 0.11/ 981 ± 81 EFC- 1.02 ± 0.10/ 829 ± 8 SEFC- 2.16 ± 0.07 /1150 ± 68 Distance— EBC- 0.48 ± .06/ 363 ± 4 SEBC- 1.08 ±. 07/495 ± 28 EFC- 0.54 ± .06/ 435 ± 23 SEFC- 1.10 ± .05/586 ± 45 | Mean Frequency— EBC- 911 ± 65 SEBC- 586 ± 77 EFC- 955 ± 33 SEFC- 664 ± 59 Duration– mean/total EBC- 1.27 ± 0.07/ 1153 ± 6 SEBC- 1.66 ± .18/961 ± 45 EFC- 1.25 ± .05/ 1192 ± 24 SEFC- 1.57 ± .07/1039 ± 53 Distance— EBC- 2.36 ± .09/ 2142 ± 70 SEBC- 2.97 ± .32/1723 ± 87 EFC- 2.31 ± .06/ 2208 ± 15 SEFC- 2.73 ± .13/1804 ± 89 | Mean Frequency— EBC- 504 ± 38 SEBC- 321 ± 75 EFC- 513 ± 26 SEFC- 352 ± 25 Duration— mean/total EBC- 1.34 ± .10/ 673 ± 9 SEBC- 1.38 ± .16/436 ± 60 EFC- 1.43 ± .09/ 730 ± 3 SEFC- 1.33 ± .03/467 ± 11 Distance— EBC- 5.67 ± .46/ 2845 ± 16 SEBC- 6.11 ± .67/1926 ± 268 EFC- 6.11 ± .42/ 3125 ± 57 SEFC- 6.02 ± 0.64/ 2112 ± 73 | Mean Frequency— EBC- 18 ± 7 SEBC- 105 ± 31 EFC- 24 ± 1 SEFC- 140 ± 14 Duration–mean/ total EBC- 0.51 ± .01/ 9 ± 1 SEBC-0.93 ± .03/ 97 ± 29 EFC-0.51 ± .03/12 ± 3 SEFC-0.98 ± .02/ 136 ± 15 Distance— EBC- 3.85 ± .01/ 70 ± 26 SEBC- 9.08 ± .38/ 952 ± 321 EFC- 3.92 ± .25/ 94 ± 9 SEFC- 9.48 ± .72/ 1329 ± 235 | | Mean Frequency—EBC- 2733 ± 142 SEBC- 1911 ± 283 EFC- 2749 ± 137 SEFC- 2014 ± 131 |
| Abdelkrim et al. [18] | Tunisian U-19 National Team (Youth) n = 6 | Frequency- All Positions- 129 ± 10 Guards-130 ± 8 Forwards- 126 ± 15 Centers- 130 ± 8 Duration- (s) All Players- 2.4 ± 0.3 Guards- 2.3 ± 0.2 Forwards- 2.4 ± 0.3 Centers- 2.6 ± 0.1 | Frequency- All Positions- 113 ± 8 Guards-113 ± 8 Forwards- 110 ± 10 Centers- 117 ± 6 Duration-(s) All Players- 2.2 ± 0.2 Guards- 2.1 ± 0.1 Forwards- 2.2 ± 0.2 Centers- 2.3 ± 0.1 | Frequency- All Positions- 97 ± 14 Guards-103 ± 11 Forwards-88 ± 5 Centers- 101 ± 19 Duration-(s) All Players- 2.3 ± 0.3 Guards- 2.1 ± 0.4 Forwards- 2.4 ± 0.2 Centers- 2.4 ± 0.4 | Frequency- All Positions- 55 ± 11 Guards-67 ± 5 Forwards-56 ± 5 Centers- 43 ± 4 Duration-(s) All Players-2.1 ± 0.2 Guards- 1.9 ± 0.2 Forwards-2.1 ± 0.1 Centers-2.2 ± 0.1 | Frequency-All Positions- 44 ± 7 Guards-41 ± 7 Forwards- 41 ± 6 Centers- 49 ± 3 | Frequency- All Positions-1050 ± 51 Guards-1103 ± 32 Forwards-1022 ± 45 Centers- 1026 ± 27 |
| Puente et al. [26] | Spanish Basketball Federation (Sub-Elite) n = 1 | Distance- (m*min) All Players- 36.4 ± 3.7 Guards- 37.7 ± 2.9 Forwards- 37.2 ± 4.6 Centers- 34.6 ± .6 | Distance- (m*min) All Players- 30.9 ± 5.9 Guards- 31.5 ± 6.9 Forwards- 32.0 ± 5.3 Centers- 29.5 ± 5.8 | Distance- (m*min) All Players- 2.3 ± 1.6 Guards- 3.1 ± 1.1 Forwards- 2.2 ± 1.9 Centers- 1.6 ± 1.6 | Distance- (m*min) All Players- 0.2 ± 0.7 Guards- 0.1 ± 0.2 Forwards- 0.5 ± 1.3 Centers- 0.0 ± 0.0 | | Distance-(m*min) All Players- 82.6 ± 7.8 Guards-85.3 ± 7.3 Forwards-86.8 ± 6.2 Centers- 76.6 ± 6.0 |
| Klusemann et al. [22] | Elite Australian Juniors (Youth) n = 13 | Frequency- Season- 255 ± 32 Tournament- 252 ± 34 | Frequency- Season-102 ± 23 Tournament- 99 ± 28 | Frequency- Season- 90 ± 17 Tournament- 82 ± 15 | Frequency- Season- 33 ± 7 Tournament- 28 ± 8 | | Frequency-Season- 809 ± 80 Tournament-758 ± 106 |

*(Continued)*

**Table 3.** (Continued)

| Study | Participants (Competition Level)n = # of comp. | Stand/Walk | Jog | Run | Sprint | Jump | All Movements |
|---|---|---|---|---|---|---|---|
| McInnes et al. [14] | Australian NBL (Elite) n = 15 | Frequency- 295 ± 54 Duration- 2.5 ± .5 | Frequency- 99 ± 36 Duration- 2.5 ± 4 | Frequency- 107 ± 27 Duration- 2.3 ± 4 | Frequency- 105 ± 52 Duration- 1.7 ± .2 | Frequency- 46 ± 12 Duration- .9 ± .1 | Frequency- 997 ± 183 |
| Abdelkrim et al. [9] | Tunisian National Team (Elite) n = 6 | Distance- (meters) 1720 ± 143 | Distance- (meters) 1870 ± 322 | | Distance-(meters) 763 ± 169 | | Distance (meters)- 7558 ± 575 |
| Ferioli et al. [32] | Italian 1st/2nd Division (Elite/Sub-Elite) n = 20 | REC Frequency- (n) Division I- 184 ± 57 Division II- 184 ± 52 Duration- (s) Division I- 1599 ± 468 Divisin II- 1757 ± 502 | LIA Frequency—(n) Division I- 306 ± 92 Division II- 296 ± 77 Duration- (s) Division I- 698 ± 213 Division II- 748 ± 200 | MIA Frequency- (n) Division I- 106 ± 31 Division II- 82 ± 34 Duration- (s) Division I- 184 ± 53 Division II- 143 ± 62 | HIA Frequency- (n) Division I- 107 ± 26 Division II- 78 ± 35 Duration- (s) Division I- 164 ± 48 Divison II- 116 ± 69 | | |

EBC = elite back-court. EFC = elite front-court. SEBC = sub-elite back-court. SEFC = sub-elite front-court. REC = recovery. LIA = low-intensity activity.

MIA = medium-intensity activity. HIA = high-intensity activity. m*min = meters per minute.

64.6 per training session whereas the $DEC_{HI}$ (n), which were classified as the total number of decelerations within the high band ($>$-3.5 m·s$^{-2}$), ranged from 1.6 to 12. When interpreting this data, it is important to acknowledge that $ACC_T$ and $DEC_T$ are qualified measures to quantify training volume, whereas $ACC_{HI}$ and $DEC_{HI}$ are quality measures of training intensity [2,10,15,43].

Remarkably, the number of $ACC_T$, $ACC_{HI}$, $DEC_T$, and $DEC_{HI}$ reported during training were considerably lower than the data found in competition settings [15,7,27]. The total volume of ACC in competition was 81 per match on average, as opposed to a mean of 38 accelerations per training session [36,40,43,47]. The total number of $ACC_{HI}$ was moderately less in training (5.6) opposed to (7.3) during match-play. This was also the case with DEC. $DEC_T$ in competition was 73.1 and the $DEC_{HI}$ 16.4, which is slightly greater than the 64.6 ($DEC_T$) and 7.4 ($DEC_{HI}$) in elite level training. The present data supports the notion that training, and match demands seem to be considerably different, at least considering the number of ACC and DEC [15]. Matching the volume and intensity of competition via training is important during certain times of the preparatory and competitive season to adequately prepare the athletes for competition. As a consequence, the data reported herein may be extremely pertinent for practitioners in regard to training reflecting the demands of match-playing, as well as modulating training load based on outputs of these variables during competition. In this context, to try and achieve similar or even greater ACC demands in training with respect to match-play, manipulating constraints such as the number of players, the duration of drills or court dimension may be a potential strategy [12,15,47]. Within this framework, Schelling and Torres [47] found that ACC load in 3vs3 and 5vs5 full court scrimmage drills was greater than 2vs2 and 4vs4 full court scrimmage drills, indeed suggesting that manipulating training variables may greatly affect the total load imposed to the players.

A study by Svilar et al. [10] reported interpositional differences in training load accelerometry data among guards, forwards, and centers. Interestingly, the authors examined load

**Table 4. Internal training load.**

| Study | Training Sessions (n=) | Participants (Competition Level) | s-RPE | Weekly TL (AU) | HR Max (BPM) | HR Average (BPM) | Max HR% | TRIMP (AU) |
|---|---|---|---|---|---|---|---|---|
| Svilar et al. [2] | n = 12 | Spanish ACB League (Elite) | 390.2±135.6 | | | | | |
| Svilar et al. [10] | n = 12 | Spanish ACB League (Elite) | Guards- 402.9 ± 151.8 Forwards- 385.5 ± 137.3 Centers- 385.1± 121.6 | | | | | |
| Ramos-Campo et al. [3] | n = 24 | Spanish ACB League (Elite) | | | 187.3 ± 10.9 | | | |
| Torres-Ronda et al. [12] | n = 15 | Spanish ACB League (Elite) | | | 5v5- 172 ± 19 4v4- 176 ± 18 3v3- 177 ± 12 2v2- 174 ± 14 | 5v5- 144 ± 17 4v4- 142 ± 15 3v3- 142 ± 15 2v2- 141 ± 15 | 5v5- 83 ± 9 4v4- 85 ± 7 3v3- 86 ± 5 2v2- 84 ± 5 | |
| Angyan et al. [25] | n = 7 | Hungarian Pro League (Elite) | | | 169 ± 5.3 | | | |
| Conte et al. [35] | n = 41 | NCAA Division I (Elite) | | Starters- 1666.2 ± 148.6 Bench- 1505.5 ± 220.8 1-game week- 1647.7 ± 251.3. 2-game week- 1423.2 ± 163.1 | | | | |
| Manzi et al. [37] | n = 200 | Italian 1st Division (Elite) | | No Game- 3334 1 Game- 2928 2 Games- 2791 | | | | |
| Heishman et al. [38] | n = 16 | NCAA Division I (Elite) | | | | | | High PL- 135.1 ±35.9 Low PL- 65.6±20.0 High Readiness- 85.3 ±19.6 Low Readiness- 104.4 ±20.1 Pre- 100.3±8.6 Post- 81.9±11 |
| Aoki et al. [39] | n = 45 | National Brazilian League (Elite) | Preseason- 442.9 ± 89.2 In-Season- 377.1 ± 68.3 | | | | | Preseason- 27.1 ± 2.1 In-Season- 21.5 ± 1.6 |
| Ferioli et al. [41] | n = 360 | Italian 1st Division/ semiprofessional (Elite/Sub-Elite) | | Pro- 5058 ± 1849 Simi-Pro- 2373 ± 488 | | | | |
| Gocentas et al. [23] | n = 42 | Semiprofessional (Sub-Elite) | | | Guards- 194 ±14 Post- 190 ± 12.7 | | | |
| Chatzinikolaoet al. [20] | n = 2 | Greek League (Elite) | | | 195 ± 6 | | | |

(*Continued*)

**Table 4.** (Continued)

| Study | Training Sessions (n=) | Participants (Competition Level) | s-RPE | Weekly TL (AU) | HR Max (BPM) | HR Average (BPM) | Max HR% | TRIMP (AU) |
|---|---|---|---|---|---|---|---|---|
| Scanlan et al. [44] | n = 44 | Australian State Level (Sub-Elite) | 47.0 ± 15.7<br>65.0 ± 17.8<br>65.0 ± 24.2<br>74.0 ± 22.7 | | | | | 31.6 ± 5.0<br>30.3 ± 6.4<br>28.8 ± 4.9<br>29.9 ± 5.4 |
| Vaquera et al. [24] | n = 26 | U-18 Spanish Juniors (Youth) | | | | | 5v5 condition (91.2 ± 4.7%. HRmax)<br>Max HR 2v2 92.7 ± 3.3% | |

s-RPE = session rate of perceived exertion. (AU) = arbitrary units. 5v5 = 5 players versus 5 players. 4v4 = 4 players versus 4 players. 3v3 = 3 players versus 3 players. 2v2 = 2 players versus 2 players.

parameters according to positional on-court roles and found that centers had a higher volume of $ACC_T$ (59.5 ± 27.1) and $ACC_{HI}$ (7.2 ± 4.8) opposed to forwards (42 ± 21.5, 5.8 ± 4.3, respectively) and guards (43.5 ± 17.5, 6.4 ± 4.4, respectively). Also, noteworthy, forwards were shown to have a high volume of $DEC_T$ (93.2 ± 35.0) and $DEC_{HI}$ (12.7 ± 8.3) compared to guards (84.7 ± 30.1, 11.9 ± 5.7) and centers (88.5 ± 30.3, 6.8 ± 4). It appears that the profiles of activity are quite different amongst positions and further research is necessary to better understand each individual profile. Still, the amount of exposures to cuts, COD, or screening actions, as well as the typical movement area of each positional role may conceivably explain such findings [6,10,12,16,27,53].

Despite the aforementioned, one must consider the limitations of accelerometry when measuring external load. Even though such technology is extremely useful, accelerometers fail to measure the metabolic demands of isometric muscle contractions during player-on-player contact due to the low velocity outputs. While these actions have very low acceleration, they potentially have very high energy demands [1,19,54]. Therefore, the physical cost of player-on-player contact loading is a component of basketball that must be examined more thoroughly in future research to more accurately quantify training and competition load.

## 6. Limitations

Some limitations should be addressed when considering the present research on training load and competition demands among different levels of basketball. Firstly, several elite leagues (e.g., NBA or ACB) do not allow for wearable technology to be used during competition which creates a gap in the literature as far as linking demands placed on the players during elite competition and how that compares to training. Secondly, when trying to investigate these variables, most sub-elite and youth teams do not have the financial means to invest in equipment to accurately quantify load during training. Finally, the limited number and sample size of youth and sub-elite studies made it difficult to conclude the precise demands of training and competition at these levels. As such, more resources need to be invested in these areas.

## 7. Conclusion

Basketball is a highly competitive team-sport that requires a cascade and flow of various movement patterns relative to the technical and tactical aspects of the sport. Examining the internal and external loads imposed on the players from both training and competition provides context for the practitioner to create an optimal training environment. Having the knowledge of

**Table 5. External training load.**

| Study | Training Sessions (n=) | Participants (Competition Level) | Acceleration | Deceleration | COD |
|---|---|---|---|---|---|
| Svilar et al. [2] | n = 300 | Spanish ACB League (Elite) | tACC- 49.1 ± 24.2 hACC- 6.5 ± 4.6 | tDEC-89.1 ± 32.2 hDEC-10.2 ± 6.8 | tCOD- 324.1 ± 116 hCOD- 21.4 ± 12.5 |
| Svilar et al. [10] | n = 208 | Spanish ACB League (Elite) | tACC- Guards- 43.5 ± 17.5 Forwards- 42 ± 21.5 Centers- 59.5 ± 27.1 hACC- Guards- 6.4 ± 4.4 Forwards- 5.8 ± 4.3 Centers- 7.2 ± 4.8 | tDEC- Guards- 84.7 ± 30.1 Forwards- 93.2 ± 35.4 Centers- 88.5 ± 30.3 hACC- Guards- 11.9 ± 57 Forwards- 12.7 ± 8.3 Centers- 6.8 ± 4.0 | tCOD- Guards- 324.8 ± 110.2 Forwards- 336.8 ± 121.4 Centers- 312.1 ± 114.8 hCOD- Guards- 23.5 ±1 2.5 Forwards- 24.7 ±1 4.5 Centers- 16.8 ± 8.6 |
| Svilar et al. [15] | n = 16 | Spanish ACB League (Elite) | tACCmin RSG- 1.92 ± 0.97 (1.78–2.06) NSG- 2.20 ± 0.76 (1.88–2.52) hACCmin RSG- 0.33 ± 0.26 (0.29–0.37). NSG- 0.25 ± 0.20 (0.17–0.34) | tDECmin RSG- 2.40 ± 1.08 (2.24–2.55) NSG- 2.95 ± 0.88 (2.58–3.23) hDECmin RSG- 0.24 ± 0.22 (0.21–0.28) NSG- 0.36 ± 0.27 (0.25–0.48) | tCODmin RSG- 10.61 ± 4.40 (9.97–11.25) NSG- 13.25 ± 3.69 (11.70–14.81) hCODmin RSG- 0.73 ± 0.46 (0.66–0.80) NSG- 0.95 ± 0.58 (0.71–1.20) |
| Vazquez-Guerrero et al. [43] | n = 33 | Spanish ACB League (Elite) | Accelerations(counts)- 1/2 court- 18.0 ± 2.4 (16.6–19.4) 1/2 court w/transition- 18.3 ± 2.8 (16.7–19.8) Full court- 16.9 ± 0.4 (16.2–17.6) hACC (counts)- 1/2 court- 1.4 ± 0.3 (1.2–1.6) 1/2 court w/transition- 1.6 ± 0.2 (1.5–1.7) Full court- 1.9 ± 0.4 (1.3–2.6) Peak Speed (m·s)- 1/2 court- 4.2 ± 0.2 (4.0–4.3) 1/2 court w/transition- 5.5 ± 0.3 (5.3–5.7) Full court- 5.0 + 0.3 (4.5–5.5) | Decelerations (counts)- 1/2 court- 17.6 ± 2.2 (16.3–18.9) 1/2 court w/transition- 17.9 ± 2.6 (16.4–19.3) Full court- 16.4 ± 0.5 (15.6–17.2) hDEC (counts)- 1/2 court- 1.1 ± 0.3 (1.0–1.3) 1/2 court w/transition- 1.4 ± 0.2 (1.3–1.5) Full court- 1. ± 0.3 (1.1–2.1) | |
| Aoki et al. [39] | n = 10 | National Brazilian League (Elite) | Peak Acceleration (m·s$^{-2}$)- Preseason- 2.2 ± 0.2 In-Season- 2.4 ± 0.2 | | |
| Scanlan et al. [44] | n = 10 | Australian State League (Sub-Elite) | Mean sprint speed (m·s) 3.77 ± 0.38 3.59 ± 0.29 3.62 ± 0.23 3.58 ± 0.30 | | |
| Schelling et al. [47] | n = 16 | Spanish ACB League (Elite) | 2v2 = 14.6 ± 2.8 3v3 = 18.7 ± 4.1 4v4 = 13.8 ± 2.5 5v5 = 17.9 ± 4.6 | | |

hACC = high-intensity acceleration. hDEC = high-intensity deceleration. tACC = total acceleration. tDEC = total deceleration. tCOD = total change of directions. hCOD = high-intensity change of directions. RSG- regular stoppage games. NSG- non-stoppage games.

the stress demands on the player during competition will help to dictate the volume and dosage of load for desirable adaptations in the player's training regimen. From the results of the present systematic review, it appears that higher-level players seem to be more efficient while moving on-court. Elite level players cover less distance, at lower average velocities, and with lower $HR_{max}$ and $HR_{ave}$ during competition. However, they seem to have greater capacities to

move at higher speed. This is likely due to a heightened sense of awareness based on the schematics of the game. Such information may provide insight into personalized testing protocols as well as training recovery strategies based on each player's response and considering mechanical and physiological loading parameters relative to competition level. Examining this holistic approach creates an ideal training environment that facilitates both technical and tactical development as it relates to the game of basketball. Future research must be dedicated to this area to provide more precise insight into the physical and interpositional demands of the sport. It is necessary to accurately and systematically assess competition demands to help determine valid training strategies that resemble match-play, considering training age, physical characteristics, and in-game role of guards, forwards, and centers. Reviewing these principals will allow priming and preparing basketball players for the rigorous of match-play demands.

## Supporting information

**S1 Checklist. PRISMA 2009 Checklist.**
(DOC)

## Acknowledgments

All contributing authors would like to acknowledge Universidad Católica San Antonio de Murcia and The Philadelphia 76ers Athlete Care Department.

## Author Contributions

**Conceptualization:** Daniel Medina Leal.

**Writing – original draft:** Adam J. Petway.

**Writing – review & editing:** Tomás T. Freitas, Julio Calleja-González, Pedro E. Alcaraz.

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
