## [Decision Letter · Decision Letter 0]

17 Dec 2019

PONE-D-19-30568

Training Load and Match-Play Demands in Basketball based on Competition Level

PLOS ONE

Dear Mr. Petway,

Thank you for submitting your manuscript to PLOS ONE. After careful consideration, we feel that it has merit but does not fully meet PLOS ONE’s publication criteria as it currently stands. Therefore, we invite you to submit a revised version of the manuscript that addresses the points raised during the review process.

---------

ACADEMIC EDITOR COMMENT:

Dear authors, thanks for submitting your work to Plos One. Find reviewer's comments below. Unfortunately, we invited 8 different reviewers but all of them declined to review except one, so the final decision about your work will be based in the sole reviewer that accepted to review, and my own. My sincere apologies for any inconveniences.

-----------

We would appreciate receiving your revised manuscript by Jan 31 2020 11:59PM. To enhance the reproducibility of your results, we recommend that if applicable you deposit your laboratory protocols in protocols.io, where a protocol can be assigned its own identifier (DOI) such that it can be cited independently in the future. For instructions see: http://journals.plos.org/plosone/s/submission-guidelines#loc-laboratory-protocols

We look forward to receiving your revised manuscript.

Kind regards,

Carlos Balsalobre-Fernández

Academic Editor

PLOS ONE

Journal Requirements:

1. We suggest you thoroughly copyedit your abstract in your manuscript for language usage, spelling, and grammar. If you do not know anyone who can help you do this, you may wish to consider employing a professional scientific editing service.  

5. We note you have included a table to which you do not refer in the text of your manuscript. Please ensure that you refer to Table 3 in your text; if accepted, production will need this reference to link the reader to the Table.

Reviewers' comments:

Reviewer's Responses to Questions

**Comments to the Author**

1. Is the manuscript technically sound, and do the data support the conclusions?

Reviewer #1: Partly

2. Has the statistical analysis been performed appropriately and rigorously? 

Reviewer #1: N/A

3. Have the authors made all data underlying the findings in their manuscript fully available?

Reviewer #1: No

4. Is the manuscript presented in an intelligible fashion and written in standard English?

Reviewer #1: No

5. Review Comments to the Author

Reviewer #1: - English grammar and spellings must be revised before a resubmission of the manuscript due to the fact that too many typos appear through the manuscript. In this sense the reviewer highlighted some of them, but I did not make an extensive review of typos.

- P2, L50, please add into brackets the specific demands (e.g., physical, technical…).

- P2, l52, please replace the term athlete with player through the text.

- P2, L54, please add after competition the different levels analysed (i.e., elite…).

- P2, L54, please write Web of Science instead of WoS.

- P2, l55-57, please replace the numbers (1), (2) and (3) with (i), (ii), (iii).

- P2, L57, please add “and” before (3).

- P2, L58, please clarify the type of study participants: …study participants into elite (xxxxxxx), sub-elite (xxxxxxx), and youth (xxxxxxxxx).

- P2, L59 and L63, add the word systematic before review.

- P2, L60, please write players instead of athlete.

- P2, L61, please write players instead of athlete.

- P2, L61, please write a full stop after athletes and start a new sentence.

- P2, L61, please delete the comma after distance.

- P2, L62, please add “than the other level players” after competition.

- P2, L66, please replace “help determine” with “provide”.

- P3, L82, please delete “of two halves, or”.

- P3, L84, please write “game-related”.

- P3, L91, what does “competition alone” mean? Please clarify the sentence and meaning of Stojanovic´s article.

- Please explain in depth the Stojanovic´s article. This reference detailed some relevant factors that affect the training load in basketball: playing period, playing position, level, geographical location and sex. In fact, the authors should justify why the current systematic review is needed, mainly when the Stojanovic´s one was published in 2017 and only a few articles were published after that date.

- P3, L92, please explain the stress variables identified in the Stojanovic´s review.

- P3, L93, please replace “Of note” with “It is important to note”.

- P3, L95, replace are with were.

- P3, L95, please write fact instead of effect.

- P4, L99-100, please write “…were the variables widely used to assign”.

- P4, L104, please replace conclusive with concluding or robust.

- P4, L106, please write “a better understanding of this…”

- P4, L109, please write …”stress that the players deal with during trainings and competitions”.

- P4, L111, please write “team sport” instead of “sport”.

- P4, L111, please write “…may be of extreme importance to ensure that players are…”.

- P4, L112, please write “…prepared for competition demands, from a fitness standpoint, in order to avoid…”.

- P4, L115, please replace dedicated to with focused on.

- P5, L121, player instead of athlete.

- P5, L123, write loads instead of load.

- P5, L125-126, please rewrite the sentence.

- P5, L127-128. The rationale of the current systematic review should be better presented and justified (as was pointed out above) highlighting the novelty of studying the different level of competition, and what does the current review add to the previous one of Stojanovic et al (2017).

- P5, L132-135, please split it into 2 sentences.

- P5, L132, please write “in the available research”.

- P5, L134-135, please write “Therefore, the aim of the present systematic review was to analyse the evidence related to…”

- P5, L140, please write focused instead of focusing.

- P6, L150, the authors should redo the search due to the fact that the first search was around 8 months ago, and not too many studies were found for sub-elite and youth levels.

- P7, L170, please write 19 years old.

- P7, L172-178, please replace the numbers (1), (2) and (3) with (i), (ii), (iii), (iv).

- P7, L174 and L178, please write and before (3) and (4), respectively.

- P7, L178, please write “a conference proceeding”.

- Quality of figures 1 and 2 must be improved.

- P8, L199, please add “the” before systematic.

- Tables 1 to 5 should include the n of matches and individual observations for each study.

- The units and symbols included in tables 1 to 5 should follow the same scale/unit of measure (m*s). In addition, the description of variables should include in all the studies mean and SD.

- Tables 1 to 5, please unify the names of the leagues to use the correct terminology. For example Spanish ACB league instead of Spanish ACB or what does Spanish Basketball Federation mean? Authors should write clearly each league name or sample used in order to avoid misunderstandings of readers and researchers.

- Results, the authors must unify the way to present the results. I suggest that mean and sd will be provided for all variables. However, the authors did this approach for some variables, but not for all.

- I suggest that the CV% would be included by authors in the analysis of each article to check the variability of their findings. This information will be useful to discuss the importance (stability) of variability among levels and then justify the large differences in some variables.

- P11, L255, please write 4,369.

- P11, L255, please write 3,722 and 6,208.

- P11, L255, please write 5,377.

- P11, L255, please write 7,558.

- P11, L255, please write 4,369.

- P11, L255, please write 5,377 and 7,558.

- P11, L257, please write In particular instead of “Of note”.

- P11, L261-262, please delete “supporting that higher-level players travel the least amount of distance during competition”.

- P11, L262-270. The authors should explain and argue some relevant factors and variables that can justify the level differences such as experience, tactical discipline, hours of training, more professional supervised practice or the control of competition.

- P12, L289, please write In fact instead Of note.

- P13, L319, garner???

- P14, L322, please replace “demonstrated” with identified or found.

- The study of Puente et al (39) should be carefully discussed and argued during results and discussion because only 1 match was studied. Then, some of the authors’ arguments need to be revised to state the limitations of sample size of some studies included in the current review.

- P16, L342, please write correctly the units of running.

- P16, L343, please write sprinting with lower case letters.

- P16, L346, please add that before there was.

- P16, L354, please write these results would instead of this would.

- P16, L355, please delete “at the”.

- P16, L 355, please replace athletes with players.

- P16, L356, please add a full stop after counterparts and clarify the last sentence.

- P16, L359, please clarify that the studies analysed time motion variables or indicators, but not time motion analysis.

- P18, L381-383, please write vs or on when explaining the game situations (e.g., 3 on 3 or 3vs3, but not 3v3).

- P18, L383, please replace games with situations.

- P21, L432, please write correctly the units of measure after >3.5.

- P22, L453-454, please write vs or on for each game situation (e.g., 3 on 3 or 3vs3, but not 3v3).

- P22, L455, please replace athletes with players.

- P22, L461-462, please justify with scientific and valid arguments the training load differences by playing position. The game play actions raised were neither studied nor real facts of the actual game play of basketball.

- P22, L468-475, please discuss these issues with valid and scientific references that support yours statements.

- Limitations, the authors should include more limitations of the current systematic review such as sample size of some studies (Puente et al) or the reduced number of studies for youth players.

- P24, L496, please replace athletes with players.

- Conclusion, L505, please add positional aspects of the sport.

- Conclusion, replace athlete with player L506, L508, L509, L510, L515, L521, and L522.

- P24, L 508, please add to before dictate.

- P24, L509, please add systematic before review.

- P25, L516, wholistic???

- Reference list should be revised in depth according to the journal guidelines: (i) journal title in italics and short title (refs 1-20 and 22-53); (ii) incomplete references without number of pages, article number of journal vol/num (refs 3 and 48); or (iii) wrong journal title ref 45 (Sports medicine instead of Springer) or ref 21.

6. PLOS authors have the option to publish the peer review history of their article (what does this mean?). If published, this will include your full peer review and any attached files.

Reviewer #1: Yes: Miguel A Gómez

---

## [Author Response · Author response to Decision Letter 0]

21 Jan 2020

Dear Reviewer,

We really appreciate the time you devoted to reading our manuscript and helping us craft an improved version. As a single reviewer we acknowledge the fact that this was a difficult process. We are pleased to clarify your concerns which we believe will improve the impact and quality of our work. Please find below our response to your observations. We have made a concerted attempt to systematically address the specific concerns raised for this revision and we have highlighted the alterations to this revision within the manuscript in yellow for your convenience. 

In advance,

King Regards

Adam J. Petway 

Reviewer #1: - English grammar and spellings must be revised before a resubmission of the manuscript due to the fact that too many typos appear through the manuscript. In this sense the reviewer highlighted some of them, but I did not make an extensive review of typos.

Author’s Response- Thank you so much for highlighting this issue. We have conducted a full revision of the English grammar to meet the standards for publication.

- P2, L50, please add into brackets the specific demands (e.g., physical, technical…). 

 Author’s Response- Thank you so much. Changed as suggested: (physiological, mechanical, technical, tactical) was added.

- P2, L52, please replace the term athlete with player through the text. 

Author’s Response- Thanks so much. Good detail. Changed as suggested. 

- P2, L54, please add after competition the different levels analysed (i.e., elite…). 

Author’s Response- Agree, Changed as suggested. Elite, Sub-Elite, and Youth. 

- P2, L54, please write Web of Science instead of WoS. 

Author’s Response- Changed as suggested.

- P2, l55-57, please replace the numbers (1), (2) and (3) with (i), (ii), (iii). 

Author’s Response- Changed as suggested.

- P2, L57, please add “and” before (3). 

Author’s Response- Changed as suggested. 

- P2, L58, please clarify the type of study participants: …study participants into elite (xxxxxxx), sub-elite (xxxxxxx), and youth (xxxxxxxxx). 

Author’s Response- Changed as suggested. We now indicate the number of studies included for each category: Elite (20), Sub-Elite (9), Youth (6).

- P2, L59 and L63, add the word systematic before review. 

Author’s Response- Changed as suggested.

- P2, L60, please write players instead of athlete. 

Author’s Response- Changed as suggested. 

- P2, L61, please write players instead of athlete. 

Author’s Response- Changed as suggested.

- P2, L61, please write a full stop after athletes and start a new sentence. 

Author’s Response- Changed as suggested.

- P2, L61, please delete the comma after distance. 

Author’s Response- Changed as suggested.

- P2, L62, please add “than the other level players” after competition. 

Author’s Response- Changed as suggested.

- P2, L66, please replace “help determine” with “provide”. 

Author’s Response- Changed as suggested. 

- P3, L82, please delete “of two halves, or”. 

Author’s Response- Changed as suggested.

- P3, L84, please write “game-related”. 

Author’s Response- Changed as suggested.

- P3, L91, what does “competition alone” mean? Please clarify the sentence and meaning of Stojanovic´s article. 

Author’s Response- “competition alone” refers to the fact that Stojanovic et al. did not examine training load and how that compared to the demands of competition. 

- Please explain in depth the Stojanovic´s article. This reference detailed some relevant factors that affect the training load in basketball: playing period, playing position, level, geographical location and sex. In fact, the authors should justify why the current systematic review is needed, mainly when the Stojanovic´s one was published in 2017 and only a few articles were published after that date. 

Author’s Response- This is a great point, thank you for bringing it up. The Stojanovic article is discussed in more depth. The fact that training load was not examined and that there is not any data reported on acceleration/deceleration demands justifies the need for the current systematic review. This is now highlighted in the text.

- P3, L92, please explain the stress variables identified in the Stojanovic´s review. 

Author’s Response- Heart rate, total distance, blood lactate and time-motion analysis are the stress variables mentioned in the article. 

- P3, L93, please replace “Of note” with “It is important to note”. 

Author’s Response- Changed as suggested.

- P3, L95, replace are with were. 

Author’s Response- Changed as suggested.

- P3, L95, please write fact instead of effect. 

Author’s Response- Changed as suggested.

- P4, L99-100, please write “…were the variables widely used to assign”. 

Author’s Response- Changed as suggested.

- P4, L104, please replace conclusive with concluding or robust. 

Author’s Response- Changed as suggested.

- P4, L106, please write “a better understanding of this…” 

Author’s Response- Changed as suggested.

- P4, L109, please write …”stress that the players deal with during trainings and competitions”. 

Author’s Response- Changed as suggested.

- P4, L111, please write “team sport” instead of “sport”. 

Author’s Response- Changed as suggested.

- P4, L111, please write “…may be of extreme importance to ensure that players are…”. 

Author’s Response- Changed as suggested.

- P4, L112, please write “…prepared for competition demands, from a fitness standpoint, in order to avoid…”. 

Author’s Response- Changed as suggested.

- P4, L115, please replace dedicated to with focused on. 

Author’s Response- Changed as suggested.

- P5, L121, player instead of athlete. 

Author’s Response- Changed as suggested.

- P5, L123, write loads instead of load. 

Author’s Response- Changed as suggested. 

- P5, L125-126, please rewrite the sentence. 

Author’s Response- Changed as suggested.

- P5, L127-128. The rationale of the current systematic review should be better presented and justified (as was pointed out above) highlighting the novelty of studying the different level of competition, and what does the current review add to the previous one of Stojanovic et al (2017). 

Author’s Response- Changed as suggested. This is a great point. We elaborated on the fact that it is helpful for coaches and sports scientists to have information about training load versus match-play demands and how that compares based on competition level. In addition, we emphasize that the present review addresses the studies that have investigated the acceleration/deceleration demands of training and competition.

- P5, L132-135, please split it into 2 sentences. 

Author’s Response- Changed as suggested. The sentence was re-arranged to make it easier for the reader to follow.

- P5, L132, please write “in the available research”. 

Author’s Response- Changed as suggested.

- P5, L134-135, please write “Therefore, the aim of the present systematic review was to analyze the evidence related to…” 

Author’s Response- Changed as suggested.

- P5, L140, please write focused instead of focusing. 

Author’s Response- Changed as suggested.

- P6, L150, the authors should redo the search due to the fact that the first search was around 8 months ago, and not too many studies were found for sub-elite and youth levels. 

Author’s Response- After conducting the search again only one new article was found that met the inclusion criteria. Vázquez-Guerrero et al. from September 2019 examined changes in physical demands between quarters. This study was conducted on youth athletes over 13 matches and, therefore, was added to the manuscript. All the other published articles were in referee, female, or wheelchair basketball and did not include training load or competition demands. 

- P7, L170, please write 19 years old. 

Author’s Response- Changed as suggested.

- P7, L172-178, please replace the numbers (1), (2) and (3) with (i), (ii), (iii), (iv). 

Author’s Response- Changed as suggested.

- P7, L174 and L178, please write and before (3) and (4), respectively. 

Author’s Response- Changed as suggested.

- P7, L178, please write “a conference proceeding”. 

Author’s Response- Changed as suggested.

- Quality of figures 1 and 2 must be improved.

- P8, L199, please add “the” before systematic. 

Author’s Response- Changed as suggested.

- Tables 1 to 5 should include the n of matches and individual observations for each study. 

Author’s Response- Number of matches and training sessions were added to the tables, as well as the competition level for training and match-play. 

- The units and symbols included in tables 1 to 5 should follow the same scale/unit of measure (m*s). In addition, the description of variables should include in all the studies mean and SD. 

Author’s Response- Changed as suggested.

- Tables 1 to 5, please unify the names of the leagues to use the correct terminology. For example Spanish ACB league instead of Spanish ACB or what does Spanish Basketball Federation mean? Authors should write clearly each league name or sample used in order to avoid misunderstandings of readers and researchers. 

Author’s Response- Changed as suggested to Spanish ACB League. Spanish Basketball Federation is the Sub-Elite second division in Spain. 

- Results, the authors must unify the way to present the results. I suggest that mean and sd will be provided for all variables. However, the authors did this approach for some variables, but not for all. 

Author’s Response- Changed as suggested. SD was added to all of the presented variables. The only case in which SD were not reported was when we presented ranges for some of the variables based on the publish data.

- I suggest that the CV% would be included by authors in the analysis of each article to check the variability of their findings. This information will be useful to discuss the importance (stability) of variability among levels and then justify the large differences in some variables. 

Author’s Response- Thanks for your comment. We agree that %CV would be a great addition to the manuscript. However, given that most articles only report mean values (and not individual values) it was not possible for us to calculate the CV for each variable. 

- P11, L255, please write 4,369. 

Author’s Response- Changed as suggested.

- P11, L255, please write 3,722 and 6,208. 

Author’s Response- Changed as suggested.

- P11, L255, please write 5,377. 

Author’s Response- Changed as suggested.

- P11, L255, please write 7,558. 

Author’s Response- Changed as suggested.

- P11, L255, please write 4,369. 

Author’s Response- Changed as suggested.

- P11, L255, please write 5,377 and 7,558. 

Author’s Response- Changed as suggested.

- P11, L257, please write In particular instead of “Of note”. 

Author’s Response- Changed as suggested.

- P11, L261-262, please delete “supporting that higher-level players travel the least amount of distance during competition”. 

Author’s Response- Deleted as suggested.

- P11, L262-270. The authors should explain and argue some relevant factors and variables that can justify the level differences such as experience, tactical discipline, hours of training, more professional supervised practice or the control of competition. 

Author’s Response- Thank you for your suggestion. We totally agree. We have re-written the sentence to bring up the point of elite athletes having a higher degree of technical and tactical economy based on training age, professional supervised practices, and higher-level coaching.

- P12, L289, please write In fact instead Of note. 

Author’s Response- Changed as suggested.

- P13, L319, garner??? 

Author’s Response- Changed garner to achieve. 

- P14, L322, please replace “demonstrated” with identified or found. 

Author’s Response- Changed as suggested.

- The study of Puente et al (39) should be carefully discussed and argued during results and discussion because only 1 match was studied. Then, some of the authors’ arguments need to be revised to state the limitations of sample size of some studies included in the current review.

Author’s Response- The limitation that only one competition was tracked was brought up as well as the fact that future research is needed.

- P16, L342, please write correctly the units of running. 

Author’s Response- All units of running were changed to (m·s). 

- P16, L343, please write sprinting with lower case letters. 

Author’s Response- Changed as suggested. 

- P16, L346, please add that before there was. 

Author’s Response- Changed as suggested.

- P16, L354, please write these results would instead of this would. 

Author’s Response- Changed as suggested.

- P16, L355, please delete “at the”. 

Author’s Response- Deleted as suggested.

- P16, L 355, please replace athletes with players. 

Author’s Response- Changed as suggested.

- P16, L356, please add a full stop after counterparts and clarify the last sentence. 

Author’s Response- Changed as suggested.

- P16, L359, please clarify that the studies analyzed time motion variables or indicators, but not time motion analysis. 

Author’s Response- Changed as suggested.

- P18, L381-383, please write vs or on when explaining the game situations (e.g., 3 on 3 or 3vs3, but not 3v3). 

Author’s Response- Changed to 5vs5 as suggested.

- P18, L383, please replace games with situations. 

Author’s Response- Changed as suggested.

- P21, L432, please write correctly the units of measure after >3.5. 

Author’s Response- Changed to (m*s-2) as suggested.

- P22, L453-454, please write vs or on for each game situation (e.g., 3 on 3 or 3vs3, but not 3v3). 

 Author’s Response- Changed to 3vs3 as suggested.

- P22, L455, please replace athletes with players. 

Author’s Response- Changed as suggested.

- P22, L461-462, please justify with scientific and valid arguments the training load differences by playing position. The game play actions raised were neither studied nor real facts of the actual game play of basketball. 

Author’s Response- The reviewer makes a compelling argument and he is right. Thank you. Interpositional demands discussed were justified by previous literature [5,6,9,13,40].

- P22, L468-475, please discuss these issues with valid and scientific references that support yours statements. 

Author’s Response- The section on interpositional demands was validated via previous work [5,6,9,13,40].

- Limitations, the authors should include more limitations of the current systematic review such as sample size of some studies (Puente et al) or the reduced number of studies for youth players. 

Author’s Response- Limitations of the Puente et al. study was discussed as well as the lack of studies within the youth and sub-elite levels.

- P24, L496, please replace athletes with players. 

Author’s Response- Changed as suggested. 

- Conclusion, L505, please add positional aspects of the sport. 

Author’s Response- Positional aspects were changed and added as future research lines due to the abovementioned reasons.

- Conclusion, replace athlete with player L506, L508, L509, L510, L515, L521, and L522. 

Author’s Response- Changed as suggested.

- P24, L 508, please add to before dictate. 

Author’s Response- Changed as suggested.

- P24, L509, please add systematic before review. 

Author’s Response- Changed as suggested.

- P25, L516, wholistic??? 

Author’s Response- Changed to global.

- Reference list should be revised in depth according to the journal guidelines: (i) journal title in italics and short title (refs 1-20 and 22-53); (ii) incomplete references without number of pages, article number of journal vol/num (refs 3 and 48); or (iii) wrong journal title ref 45 (Sports medicine instead of Springer) or ref 21. 

Author’s Response- Italics and short titles were corrected as well as the vol and title for 3, 48, 45, and 21.

---

## [Editor Report · Decision Letter 1]

3 Feb 2020

Training Load and Match-Play Demands in Basketball based on Competition Level: A Systematic Review

PONE-D-19-30568R1

Dear Dr. Petway,

We are pleased to inform you that your manuscript has been judged scientifically suitable for publication and will be formally accepted for publication once it complies with all outstanding technical requirements.

With kind regards,

Carlos Balsalobre-Fernández

Academic Editor

PLOS ONE
---

## [Editor Report · Acceptance letter]

18 Feb 2020

PONE-D-19-30568R1 

Training Load and Match-Play Demands in Basketball based on Competition Level: A Systematic Review 

Dear Dr. Petway:

I am pleased to inform you that your manuscript has been deemed suitable for publication in PLOS ONE. Congratulations! Your manuscript is now with our production department. 

With kind regards,

on behalf of

Dr. Carlos Balsalobre-Fernández 

Academic Editor

PLOS ONE